# Sequential Bayesian Experimental Design with Variable Cost Structure

**Sue Zheng**
CSAIL MIT
szheng@csail.mit.edu

**David S. Hayden**
CSAIL MIT
dshayden@csail.mit.edu

**Jason Pacheco**
University of Arizona
pachecoj@cs.arizona.edu

**John W. Fisher III**
CSAIL MIT
fisher@csail.mit.edu

## Abstract

Mutual information (MI) is a commonly adopted utility function in Bayesian optimal experimental design (BOED). While theoretically appealing, MI evaluation poses a significant computational burden for most real world applications. As a result, many algorithms utilize MI bounds as proxies that lack regret-style guarantees. Here, we utilize two-sided bounds to provide such guarantees. Bounds are successively refined/tightened through additional computation until a desired guarantee is achieved. We consider the problem of adaptively allocating computational resources in BOED. Our approach achieves the same guarantee as existing methods, but with fewer evaluations of the costly MI reward. We adapt knapsack optimization of best arm identification problems, with important differences that impact overall algorithm design and performance. First, observations of MI rewards are biased. Second, evaluating experiments incurs shared costs amongst all experiments (posterior sampling) in addition to per-experiment costs that may vary with increasing evaluation. We propose and demonstrate an algorithm that accounts for these variable costs in the refinement decision.

## 1 Introduction

In many analysis problems the data collection process is subject to resource limitations. These limitations arise in a variety of ways, including limits on explicit costs, selection size, energy expenditures, time expenditures, and limits on *computation*. The framework of Bayesian optimal experimental design (BOED) judiciously allocates limited resources by identifying a sequence of designs that are maximally informative about a quantity of interest. This optimization of resources is not unique to the design of experiments. Indeed, the decision making task which underlies BOED is analogous to that of other domains, owing to its significance: sensor planning [14, 27], active learning [8, 25], multi-armed bandit best arm identification [19], among others.

Various measures exist to assess the information content of a design. Following classical foundations in BOED [3, 20, 2], this work considers mutual information (MI) as a design utility. MI quantifies the expected reduction in posterior uncertainty provided by an experiment, and offers a number of advantages to alternative information measures. Most importantly, it is invariant to deterministic mappings under additive noise [18]. By contrast, classical measures based on Fisher information (*e.g.* A-, D-, E-, etc. optimal design) have function-specific biases. Indeed, MI equates to D-optimal design for normal linear models in the classic formulation of BOED [6, 4]. All information measures, however, pose unique computational challenges for even moderately complex latent variable models.

Owing to the intractability of exact MI evaluation, many practical BOED implementations use estimates or bounds to serve as *proxies* for the information reward. Sample-based estimates of MI can be computationally prohibitive since drawing samples relies on inference to incorporate experimental results from the previous stage. Naïve estimates require many samples to achieve sufficient accuracy [24]. While there are approaches that mitigate this issue, *e.g.* [29], sample-based approximations quickly become challenging for problems with large action sets or high dimension. Alternatives to sample-based estimates include the use of variational inference combined with variational bounds on MI [22] and sample-based variational upper or lower bounds on MI [11, 12, 7]. However, the impact on performance is unknown for these MI proxies. For example, it is possible that experimental choices may be dominated by tighter bounds in favor of higher rewards. Consequently, the resulting selection may be arbitrarily bad as compared to the optimal experiment.

In contrast to existing BOED methods that use lower *or* upper bounds as plug-in estimates of MI, this work proposes using two-sided bounds to provide guarantees on performance relative to optimal (*e.g.* regret). Guarantees allow us to identify when low-fidelity bounds suffice to identify high-quality designs, leading to computational savings. Furthermore, they allow for an iterative approach that targets a specified performance level by tightening/refining bounds with additional computation. We adaptively allocate resources to refine the bounds on specific designs in a manner sensitive to the computational costs of reward evaluation. Unlike previous methods, this work considers both the costs and value associated with iterative refinement of MI bounds. This cost-aware allocation allows us to trade off computational resources with fidelity of the reward evaluation. By formulating resource allocation as a knapsack problem, we optimize the trade-off between computation and tightness of the bounds while achieving guarantees on the optimality of the selection.

**Contributions:** This work makes a number of contributions to the field of experimental design: (1) We incorporate both lower and upper bounds on the intractable MI utility into BOED to provide performance guarantees relative to the optimal design and show how they can be maintained with minimal additional computation. (2) We introduce a cost-sensitive sequential optimization that iteratively refines bounds to target a desired performance level. (3) We formulate a greedy knapsack optimization that elegantly trades off performance for computation. (4) Finally, we evaluate our BOED method against adaptive allocation techniques in Gaussian models as well as the challenging problem of multi-target tracking.

## 2 Sequential Bayesian Experimental Design

Sequential Bayesian optimal experimental design (BOED) seeks to identify a series of experiments that yields the most information about an unknown quantity of interest, $x$. At time $t$, observations $y_t$ are driven by the choice of experiment design $a_t \in \{1, \ldots, A\}$. Intuitively, the design $a_t$ parameterizes a likelihood model $p_{a_t}(y_t \mid x)$. Assuming measurements are conditionally independent, the posterior given a history of past observation/design pairs $\mathcal{D}_T = \{(y_t, a_t)\}_{t=1}^T$ is given by,

$$p(x \mid \mathcal{D}_T) \propto p(x) \prod_{t=1}^T p_{a_t}(y_t \mid x). \tag{1}$$

At each time $t$ sequential Bayesian design selects the experiment that maximally reduces a measure of posterior uncertainty. In particular, we maximize the mutual information (MI) as in [20]:

$$a_t^* = \arg\max_a I_a(X; Y_t \mid \mathcal{D}_{t-1}) \triangleq \mathrm{H}(X \mid \mathcal{D}_{t-1}) - \mathrm{H}_a(X \mid Y_t, \mathcal{D}_{t-1}), \tag{2}$$

where $\mathrm{H}_a(X \mid Y, \mathcal{D}) \triangleq - \int p(x, y \mid \mathcal{D}) \log p_a(x \mid y, \mathcal{D}) \, \mathrm{d}x \, \mathrm{d}y$ is the conditional differential entropy, and $\mathrm{H}(X \mid \mathcal{D}) \triangleq - \int p(x \mid \mathcal{D}) \log p(x \mid \mathcal{D}) \, \mathrm{d}x$ the posterior entropy. The sequential *greedy* optimization in Eqn. (2), while myopic, avoids the complexity associated with finding an optimal design policy. Even then, greedy sequential experimental design is complicated since MI lacks a closed-form solution in all but trivial cases. This requires the use of proxies for the MI reward, which we discuss next. The procedure for sequential design with a sample-based estimator is outlined in Alg. 1.

---

**Algorithm 1** Sequential Bayesian Experiment Design

---

**Input:** max samples $N_{\max}$
Initialize data: $\mathcal{D} \leftarrow \emptyset$
**for** $t = 1$ to $T$ **do**
    Draw $N_{\max}$ samples: $\{x_n, y_n\}_{n=1}^{N_{\max}} \sim p(x, y \mid \mathcal{D})$
    **for** $a = 1$ to $|\mathcal{A}|$ **do**
        Approximate MI from samples: $\hat{I}_a(X; Y \mid \mathcal{D})$
    **end for**
    Maximize proxy: $\hat{a}^* \leftarrow \arg\max_a \hat{I}_a$
    **Return:** $\hat{a}^*$
    Run experiment, collect data: $y_a \sim p_{\hat{a}^*}(y)$
    $\mathcal{D} \leftarrow \mathcal{D} \cup y_a$
**end for**

---

## 2.1 Calculating and Bounding Mutual Information

The MI measure in Eqn. (2) is notoriously difficult to compute [21]. To highlight the difficulty we drop explicit dependence on time and design for brevity. The conditional entropy is given by,

$$\mathrm{H}(X \mid Y, \mathcal{D}) = \mathbb{E}\left[ -\log \frac{p(x, y \mid \mathcal{D})}{p(y \mid \mathcal{D})} \right]. \tag{3}$$

Evaluating Eqn. (3) requires pointwise evaluation of the posterior predictive distribution $p(y \mid \mathcal{D}) = \int p(x \mid \mathcal{D}) p(y \mid x) \, \mathrm{d}x$, a posterior expectation that typically lacks a closed-form. MI can be bounded using any unbiased estimator $\mathbb{E}[\hat{p}(y)] = p(y \mid \mathcal{D})$ of the posterior predictive density. By Jensen's inequality we have $\mathbb{E}[\log \hat{p}(y)] \leq \log p(y \mid \mathcal{D})$. Given samples $\{x_n, y_n\}_{n=1}^N \sim p(x, y \mid \mathcal{D})$ the nested Monte Carlo estimator yields the known lower [26, 12] and upper [23, 11] bounds $(l, u)$,

$$l = \frac{1}{N} \sum_{n=1}^N \log \frac{p(y_n|x_n)}{\frac{1}{N}\sum_{m=1}^N p(y_n|x_m)}, \quad u = \frac{1}{N} \sum_{n=1}^N \log \frac{p(y_n|x_n)}{\frac{1}{N-1}\sum_{m \neq n} p(y_n|x_m)}, \tag{4}$$

which hold in expectation: $I(X; Y) \geq \mathbb{E}[l]$ and $I(X; Y) \leq \mathbb{E}[u]$. Because the bounds differ by one sample-point in the denominator, minimal additional computation is needed to obtain *both* bounds as compared to a single bound, a property we will exploit in Sec. 3.

## 2.2 Challenges of Proxy-Based BOED

Given the difficulty of computing MI, a lower $l$ or upper $u$ bound is commonly used as a proxy in BOED [29, 24, 11, 23]. Designs are sequentially chosen by maximizing the chosen proxy $\hat{I}_a$, as in: $\hat{a}_t^* = \arg\max_a \hat{I}_a$. The bounds in Eqn. (4) become arbitrarily tight as $N \to \infty$, and can therefore be refined with additional computation (*i.e.* drawing more samples). However, the use of single-sided bounds as an MI proxy poses several problems. In particular, it is impossible to ensure performance guarantees on information gain as the proxy may reflect the bound gap rather than the MI reward. As seen in Fig. 1, two proxies using the same budget have significant differences in quality that also depends on the problem structure; a single budget specification for all cases yields wasted computation or poor performance. Additionally, bound refinement is limited to the naïve allocation of samples without knowledge of expected improvement in the chosen design quality.

## 3 BOED with Cost-Sensitive Iterative Refinement

In this section we outline our framework for Bayesian optimal experimental design with iterative refinement (BOEDIR) of the bounds. It differs from the standard framework of Alg. 1 in the inclusion of two-sided bounds, the provision of a performance guarantee relative to optimal, and the incremental allocation of computation to tighten bounds of select designs. BOEDIR admits any bounds on MI and allows for specification of a performance metric for termination.

The BOEDIR framework is summarized in Alg. 2. Let $\mathrm{SelectRefine}$ be any algorithm that returns $\mathcal{R}$, a set of designs for refinement. First, a minimal amount of computation is used to (loosely) bound every design. This may suffice to exclude some designs from further computational resources.

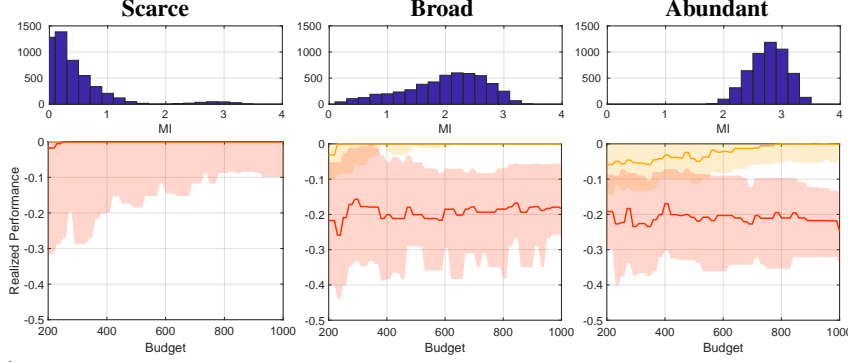

Figure 1: **Realized Performance Across Motifs in BOED.** *Top-row:* Three distributions over MI rewards characterize different problem structures or "motifs". *Bottom-row:* Median and quartiles performance (information gain relative to optimal) under proxy-based BOED framework for two proxies under various budgets: lower bound (`bed-lb`) and upper bound (`bed-ub`). Motifs are: *scarce* mostly uninformative *(left)*, *broad* similarly informative *(middle)*, and *abundant* mostly informative *(right)*. Performance varies greatly depending on the proxy used and the problem structure, leading to wasted compute cycles when the specified budget is too large (*scarce* with `bed-lb`) or poor performance when not large enough (*abundant* with `bed-ub`).

---

**Algorithm 2** BOED with Iterative Refinement of Bounds

---

**Input:** target $\tau$, metric $g(l, u)$, max resources $C_{\max}$
Initialize: data $D \leftarrow \emptyset$
**for** $t = 1$ to $T$ **do**
    Initialize: perfMet = false, cost $C = 0$, refinement set $\mathcal{R} = \mathcal{A}$
    **repeat**
        **for** $a \in \mathcal{R}$ **do**
            Refine bounds on MI: $l_a$, $u_a$                              // Eqn. (4)
            Update cost: $C \leftarrow C + c_a$                          // Eqn. (8)
        **end for**
        Update refinement set: $\mathcal{R} \leftarrow \text{SelectRefine}(\{l_a, u_a, c_a\}_{a=1}^{|\mathcal{A}|})$      // Eqn. (7)
        Evaluate guarantees: $\{g_a = g(l_a, u_{\backslash a})\}_{a \in \mathcal{A}}$      // *e.g.* Eqn. (5)
        Highest guarantee: $g^* \leftarrow \max_a g_a$, $\hat{a}^* \leftarrow \arg\max_a g_a$
        Test requirement: perfMet $\leftarrow g^* \geq \tau$
    **until** perfMet is true or $C > C_{\max}$
    **Return:** $g^*$, $\hat{a}^*$
    Run experiment, collect data: $D \leftarrow D \cup y_{\hat{a}^*}$
**end for**

---

We then iteratively select design(s) to refine ($\mathcal{R} \leftarrow \text{SelectRefine}$), expend computation to tighten their bounds $(l_a, u_a)_{a \in \mathcal{R}}$, and update the performance guarantee. This continues until the target performance is met or the budget on resources is exhausted. We propose a cost-aware algorithm for SelectRefine based on the greedy knapsack algorithm and compare against two bandit-style alternatives in our experiments. Mechanisms for bound refinement vary depending on the form of the bound: drawing additional samples can tighten sample-based bounds while performing additional rounds of gradient descent or broadening the variational class can tighten variational bounds. Here, we focus discussion on the bounds of Eqn. (4) but provide further discussion of alternative bounds and their refinement and costs in the Supplemental. We now describe the performance guarantees, costs that arise with bound refinement, and refinement selection in greater detail.

**Performance Guarantees** Let $g(I_a, I_{a^*})$ be a metric that measures the performance of a design $a$ relative to the optimal design $a^*$. Performance metrics are increasing in $I_a$ and decreasing in $I_{a^*}$. Relevant metrics depend on the problem at hand so we allow for arbitrary $g(I_a, I_{a^*})$ but will develop discussion for two common metrics, *negative* regret and relative optimality:

$$g_{\text{regret}}(I_a, I_{a^*}) = I_a - I_{a^*} \qquad\qquad g_{\text{pct}}(I_a, I_{a^*}) = \frac{I_a}{I_{a^*}}. \qquad (5)$$

While we cannot evaluate the performance metric since we do not know $I_a$ and $I_{a^*}$, we can bound it using MI lower and upper bounds $(l_a, u_a)_{a \in \mathcal{A}}$ to obtain a performance guarantee *for every design*,

$$g_{\text{regret}}(I_a, I_{a^*}) \geq g_{\text{regret}}(l_a, u_{\backslash a}) \qquad\qquad g_{\text{pct}}(I_a, I_{a^*}) \geq g_{\text{pct}}(l_a, u_{\backslash a}) \qquad (6)$$

where $u_{\backslash a} = \max_{k \in \mathcal{A} \backslash a} u_k$ is the highest upper bound, excluding design $a$. Thus, by having upper and lower bounds on designs we can ensure sufficiently high quality selections.

## 3.1 Knapsack Refinement

To identify designs that rapidly approach the target performance while using as few resources as possible, we adapt the greedy knapsack algorithm [9]. Using the knapsack greedy heuristic, our approach allocates computation towards the design with the highest value-to-cost ratio, known as the *marginal utility*. Specifically, the value of refining design $a$ is the improvement $\Delta_a g^*$ it provides on the highest performance guarantee $g^* = \max_a g_a$. Costs $c_a$ are associated with refinement of each bound and can vary by experimental design or even with degree of refinement. We select the design with highest marginal utility for bound refinement:

$$\mathcal{R} = \arg\max_a \frac{\Delta_a g^*}{c_a}. \tag{7}$$

**Costs for Sample-based Bounds** We maintain a set of joint posterior samples $\mathcal{X}$ and observation samples $\mathcal{Y}_a$. When design $a$ is selected for refinement, an additional $\eta$ observation samples are drawn, each with cost $c_y$. The updated sample count is $N = |\mathcal{Y}_a| + \eta$. Posterior samples each with cost $c_p$ represent a *shared* cost as they can be used for all design bounds. They are drawn only when the updated observation sample count exceeds the existing posterior sample count, $N \geq |\mathcal{X}|$. Lastly, given the nature of the nested bound estimates of Eqn. (4), the evaluation of bounds depends quadratically on the number of samples $N$ with cost $c_{\text{bound}} = d_0 + N d_1 + N^2 d_2$. The net cost is

$$c_a = \eta c_p \cdot \mathbb{1}_{N \geq |\mathcal{X}|} + \eta c_y + c_{\text{bound}}. \tag{8}$$

While these parameters $c_y, c_p, d_0, d_1, d_2$ are dependent on the problem, inference procedure, and other factors, they are usually readily estimated using any number of methods for measuring code performance, including functions that measure wall time. The coefficients of the bounding function can be estimated by a quadratic fit to timing measurements at various sample sizes. Alternatively, one could learn these parameters online; measuring and adaptively estimating the computational cost adds little computation.

**Value of Refinement** Let $g_a = g(l_a, u_{\backslash a})$ be the performance guarantee on design $a$ for the metric of interest (*e.g.* negative regret). We are interested in refining bounds to improve the highest guarantee $g^* \triangleq \max_a g_a$. For computational simplicity, we use a first-order approximation to the change in $g^*$ assuming a nominal update to lower/upper bounds $(\Delta l_a, \Delta u_a)$ and expand using the chain rule:

$$\Delta_a g^* = \frac{\partial g^*}{\partial l_a} \Delta l_a + \frac{\partial g^*}{\partial u_a} \Delta u_a = \sum_b \frac{\partial g^*}{\partial g_b} \frac{\partial g_b}{\partial l_a} \Delta l_a + \sum_b \frac{\partial g^*}{\partial g_b} \frac{\partial g_b}{\partial u_a} \Delta u_a. \tag{9}$$

Let us consider $\frac{\partial g_b}{\partial l_a}$ and $\frac{\partial g_b}{\partial u_a}$ in detail. Recall the design's guarantee $g_a = g(l_a, u_{\backslash a})$ depends on its own lower bound $l_a$ and the maximum upper bound of *other designs* $u_{\backslash a} = \max_{k \in \mathcal{A} \backslash a} u_k$. Thus, tightening $l_a$ only improves $g_a$ (*i.e.* for $b \neq a$, $\frac{\partial g_b}{\partial l_a} = 0$) and tightening $u_a$ potentially updates $g_b$ through $u_{\backslash b}$ for $b \neq a$ (*i.e.* $\frac{\partial g_a}{\partial u_a} = 0$ and $\frac{\partial g_b}{\partial u_a} = \frac{\partial g_b}{\partial u_{\backslash b}} \frac{\partial u_{\backslash b}}{\partial u_a}$). Substitution into Eqn. (9) gives,

$$\Delta_a g^* = \frac{\partial g^*}{\partial g_a} \frac{\partial g_a}{\partial l_a} \Delta l_a + \sum_{b \neq a} \frac{\partial g^*}{\partial g_b} \frac{\partial g_b}{\partial u_{\backslash b}} \frac{\partial u_{\backslash b}}{\partial u_a} \Delta u_a. \tag{10}$$

Both $g^* = \max_a g_a$ and $u_{\backslash b} = \max_{k \in \mathcal{A} \backslash b} u_k$ are discontinuous maximum functions which have poor behavior in a first-order approximation. Consider the following example where design $a$ has guarantee *close* to the highest, $g_a = g^* - \epsilon$. Updating its lower bound can cause it to overtake the current highest guarantee and improve $g^*$. However, the gradient of $\max$ gives $\frac{\partial g^*}{\partial g_a} = 0$, incorrectly suggesting no impact on $g^*$ from updating $l_a$. We instead evaluate partials for $\max$ based on a standard smooth approximation to $\max$: LogSumExp (LSE) which yields softmax gradient, $\frac{\partial g^*}{\partial g_a} = \frac{\exp(g_a)}{\sum_{a'} \exp(g_{a'})}$. The remaining partials $\frac{\partial g_a}{\partial l_a}, \frac{\partial g_a}{\partial u_{\backslash a}}$ depend on the choice of metric. For the aforementioned metrics, they are $\frac{\partial g_{\text{regret},a}}{\partial l_a} = 1, \frac{\partial g_{\text{regret},a}}{\partial u_{\backslash a}} = -1$ and $\frac{\partial g_{\text{pct},a}}{\partial l_a} = \frac{1}{u_{\backslash a}}, \frac{\partial g_{\text{pct},a}}{\partial u_{\backslash a}} = -\frac{l_a}{u_{\backslash a}^2}$. Lastly, we specify the nominal update to the upper and lower bounds as a fraction $\gamma_{a,u}, \gamma_{a,l}$, respectively, of the difference between the existing upper and lower bounds: $\Delta u_a = \gamma_{a,u} * (u_a - l_a)$. We adaptively estimate these fractions from observed updates to the bounds using an exponentially weighted moving average.

## 3.2 Relation to Best Arm Identification Multi-armed Bandits

Optimal design with consideration to limitations in resources has been studied in best-arm identification (BAI) problems for multi-armed bandits (MAB) [5]. This setting optimizes simple regret rather than cumulative, thereby decoupling resources from rewards and eliminating the exploitation/exploration trade-off considerations of standard MAB [15]. Key differences separate BAI from our setting. First, BAI costs are typically treated as constant and uniform across arms, with some exceptions [13, 28]. Second, BAI typically assumes unbiased observation of the reward (one notable exception being [16]). Third, we allow for computational expenditures shared across arms (i.e., inference to sample latent parameters). Despite the different assumptions, we can still utilize BAI algorithms within our framework as a refinement selection algorithm with the caveat that existing guarantees from BAI do not hold in our setting. We will evaluate two styles of BAI algorithms that achieve optimal sample complexity under fixed confidence [15]: action elimination (AE) [10] and lower upper confidence bound (LUCB) [17]. We do not consider UCB [1] which, when applied to BAI problems, can be overly exploitative [15].

## 4 Experimental Results

We evaluate performance in a Gaussian Markov Random Field (MRF) model which allows for exact evaluation of MI. In this setting, BOEDIR with refinement selection based on marginal utility (`mu`) performs comparably to non-iterative proxy-based BOED that utilizes either the lower (`bed-lb`) or upper bound (`bed-ub`) in Eqn. 4 as a proxy. Within the iterative framework, we additionally compare `mu` against two alternative strategies for refinement selection: action elimination (`ae`) and lower upper confidence bound (`lucb`). AE maintains a set of potential optimal arms and uniformly refines the set. We define that set as designs with upper bound greater than the highest lower bound $\mathcal{R} = \{a : u_a > \max_{a'} l_{a'}\}$. LUCB, a variant on UCB, selects two arms for refinement: the design with the highest performance guarantee $\hat{a}^* = \arg\max_{a' \in \mathcal{A}} g_{a'}$ and the design of the remaining with highest upper bound $\bar{a} = \arg\max_{a' \in \mathcal{A} \setminus \hat{a}^*} u_{a'}$, $\mathcal{R} = \{\hat{a}^*, \bar{a}\}$. Finally, we consider the more challenging problem of track ambiguity in multi-object tracking (MOT) where we seek informative annotations to resolve the ambiguities. We obtain computational savings over an alternative refinement approach and achieve performance comparably to a proxy-based approach utilizing the full computation budget.

## 4.1 Gaussian MRF

For analysis purposes, we evaluate performance on a tree-structured Gaussian MRF. In this setting, MI can be computed exactly, thereby allowing for comparison of realized performance. We demonstrate our approach under three different *motifs*, or problem structures, characterized by their distributions over MI shown in Fig. 1 and at various relative costs for posterior inference since this cost is highly implementation and problem dependent. We consider posterior costs $c_p$ that are $100, 10, 1$, and $0.1\times$ the cost $c_y$ of sampling $y_a$ while holding $c_y$ fixed.

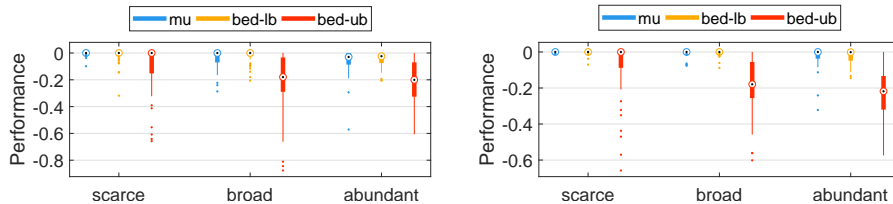

Figure 2: **Performance Against Proxy-based Baseline.** Median and quartiles from 50 random trials of realized performance for three motifs. We compare our approach (`mu`) against two proxies, `bed-lb` and `bed-ub`, in the standard BOED framework subject to the same budget, $C_{\max} = 500$ (*left*) and $C_{\max} = 1000$ (*right*).

Consider a tree-structured GMRF $\mathcal{G} = (\mathcal{E}, \mathcal{V})$ with edges $\mathcal{E}$, nodes $\mathcal{V} = \mathcal{V}_x \cup \mathcal{V}_y$, and joint probability, $\mathcal{N}(x \mid m_x, \Sigma_x) \prod_{(s,a) \in \mathcal{E}: s \in \mathcal{V}_x, a \in \mathcal{V}_y} \mathcal{N}(y_a \mid C_a x_s, \sigma_a^2)$. Latent nodes $x_s$ are 2D Gaussian random variables and observations $y_a$ are scalar. The likelihood model is defined over a set of random linear projections with parameters $\{C_a\}_{a=1}^A$ and noise variance $\{\sigma_a^2\}_{a=1}^A$. At each stage of design the algorithm must choose the projection maximizing $I_a(X; Y)$.

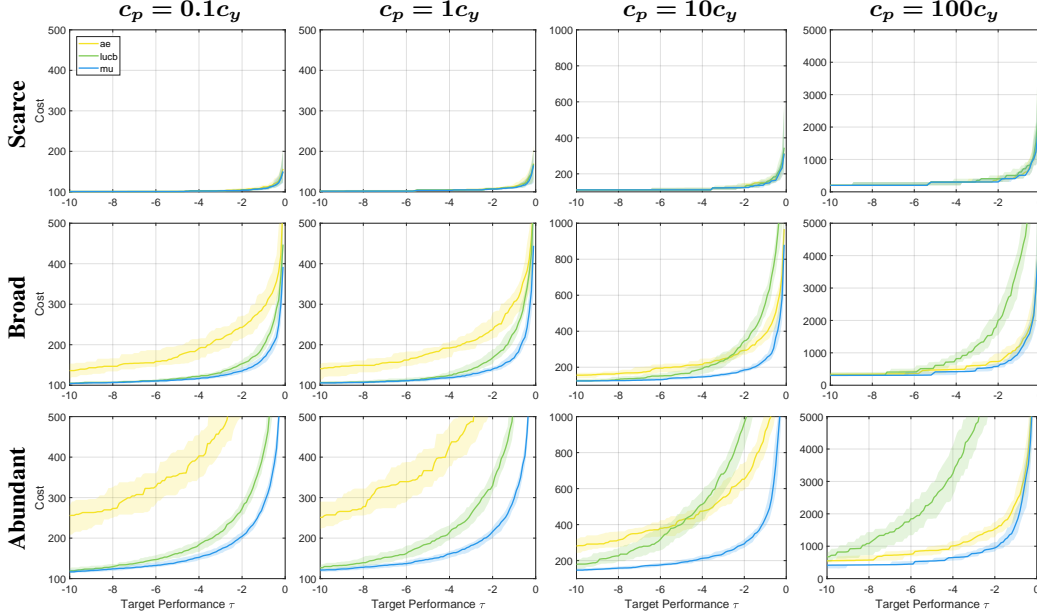

**Figure 3: Cost Vs Targeted Performance as Overhead Cost Increases for Three Motifs.** Median and quartiles from 50 random trials of cost at each target performance level for different distributions of MI rewards. Posterior sampling costs vary from [0.1, 1, 10 100] (*left-right*). Motifs are: *scarce* mostly uninformative (*top-row*), *broad* similarly informative (*middle-row*), and *abundant* mostly informative (*bottom-row*). Baselines `ae` and `lucb` perform well at low and high posterior costs respectively whereas `mu` performs well across cost ratios. All require equally little computation in *scarce* as loose bounds suffice to identify many designs as poor. As more designs yield high information, additional computation is required to refine bounds sufficiently to achieve the target performance guarantee.

We generate random trees with $|\mathcal{V}_x| = 50$ nodes, each latent node having two randomly generated candidate projection operators. This results in $|\mathcal{V}_y| = 100$ total candidate experiments to choose from. We demonstrate the performance guarantee/computation trade-off using the BOEDIR framework under different problem structures and posterior cost ratios. We do not have experiments explicitly analyzing the impact of increasing the size of the design set, but expect our approach to yield greater savings (with respect to baselines) as allocating resources to promising designs is increasingly important.

**Comparison to Standard BOED Framework** We evaluate performance when subject to a budget constraint rather than a target performance guarantee as in the standard BOED framework. Realized performance under the lower bound proxy (`bed-lb`), upper bound proxy (`bed-ub`), and our cost-sensitive refinement approach (`mu`) is shown in Fig. 2. Performance is measured as *negative* regret; optimal performance is zero, reflecting no loss. `mu` performs similar well to `bed-lb` while also providing a performance guarantee. Ostensibly, one could simply select the better proxy `bed-lb` for use in standard BOED. However, it is often challenging to ascertain the quality of the proxy to make this determination; exact evaluation of MI is possible in few cases such as the Gaussian MRF. Without awareness of the better proxy, `mu` incorporates both bounds to obtain performance close to `bed-lb` across motifs while also providing assurances on the quality of the selected design. As shown in the Supplemental, we find this behavior to persist across budget settings and relative posterior costs.

**Comparison of Refinement Selection Algorithms in BOEDIR** We demonstrate the dependence of cost required by each refinement selection algorithm on target performance levels for three motifs in Fig. 3. We again use negative regret as the performance metric such that $\tau = 0$ corresponds to optimal performance. As the target performance guarantee increases, all algorithms require additional computational cost to provide the guarantee; cost increases dramatically as the target guarantee nears optimal. Significant computational savings can be obtained by relaxing the target guarantee.

Our algorithm requires lower or equivalent cost compared to the baselines across all variants in relative posterior costs, motifs, and target performance guarantee. Comparison across relative posterior costs

reveals that the baseline BAI algorithms have cost regimes wherein they perform effectively. AE, by uniformly refining all arms in the set before revisiting any, is inherently conservative towards incurring overhead. We find that its performance, relative to other algorithms, improves as overhead becomes more expensive. LUCB on the other hand, hones in on the few arms that appear optimal. This requires more posterior samples to obtain better quality bounds on those arms. When posterior inference is cheap, an uncommon occurrence in real world problems, this approach does well. Our cost-sensitive approach performs well in a range of cost regimes.

Evaluation of the algorithm on different motifs reveals how costs depend on the underlying distribution of MI. In *scarce*, all perform similarly well since many poor actions are readily identified as such even with loose bounds. Only a few actions merit refinement and algorithms perform similarly as there is little need to be clever in the refinement selection. *Broad* has more actions that are highly rewarding, thus additional computation is required to discern a better action from a merely good one. Lastly, with *abundant* high reward options, all algorithms require further computation to identify a sufficiently good one in this challenging motif. Our algorithm sees increasing gains over alternatives on more challenging motifs, the situations where cost reduction is most necessary.

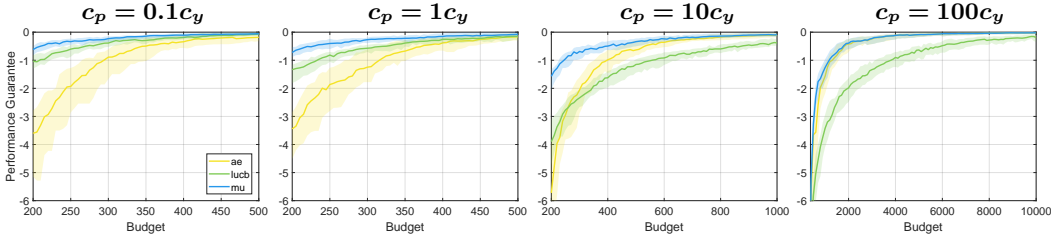

Figure 4: **Guarantee Vs Budget for 'Broad'.** Median and quartiles from 50 random trials for performance guarantee at each budget for *broad* spread in rewards. Posterior sampling cost ratio varies from [0.1, 1, 10 100] (*left-right*). Our algorithm consistently provides higher guarantees across various budgets and relative costs.

One can also consider a fixed-budget formulation wherein we optimize the performance guarantee subject to expending the full computational budget. The dependence of performance guarantee on budget for the *broad* motif is shown in Fig. 4. The results for the other motifs are given in the Supplemental since the same trends hold in this formulation and mirror the results when we target a performance guarantee: mu performs better or as well as the BAI baselines in all variants of posterior cost ratio and budget settings except in the simplest motif wherein all algorithms perform similarly.

## 4.2 Multi-object Tracking

Data association is a challenging problem that arises in a variety of problems including multi-object tracking (MOT). When formulated as the unique assignment of $K$ measurements to $K$ objects over $T$ time points, this yields $(K!)^T$ possible assignments $z$. Marginalization over these assignments to obtain a posterior distribution over joint target states, $x$ quickly becomes infeasible. Instead, methods such as MCMC sampling on both target states and assignments are used. Analysis often reveals multiple modes in the posterior since target ambiguity results from objects becoming kinematically close then separating. We consider the use of labeled associations from an annotator to resolve ambiguities. Here, we consider an annotator who reports whether two sensor observations arise from the same underlying object with error probability $p_a$ and seek the annotation which maximizes information of the latent target states. Details for this model are provided in the Supplemental.

We evaluate on a scenario with $K = 3$ targets that become kinematically confused in "entanglement" events, shown in Figure 6 (*left*). There are $2! * 2! * 3! = 24$ possible outcomes depending on the alignment of entering to exiting tracks at each entanglement. These outcomes correspond to modes in the posterior. Because we cannot evaluate the posterior entropy of the track state, we focus on modes in the target state posterior to ascertain ambiguity. Following each annotation, we incorporate the new annotation data and draw 3000 posterior samples. Knowing the major 24 posterior modes, we map each sample to one of these. The distribution over modes reflects the degree of track ambiguity; a uniform distribution reflects high ambiguity. We calculate the discrete entropy of this empirical distribution to quantify the ambiguity and repeat for 50 random trials. Because this problem setting exhibits high relative posterior sampling costs - 2000× for single-chain MCMC and 125× for 16 parallelized chains, we only compare against ae which performs well in this cost regime.

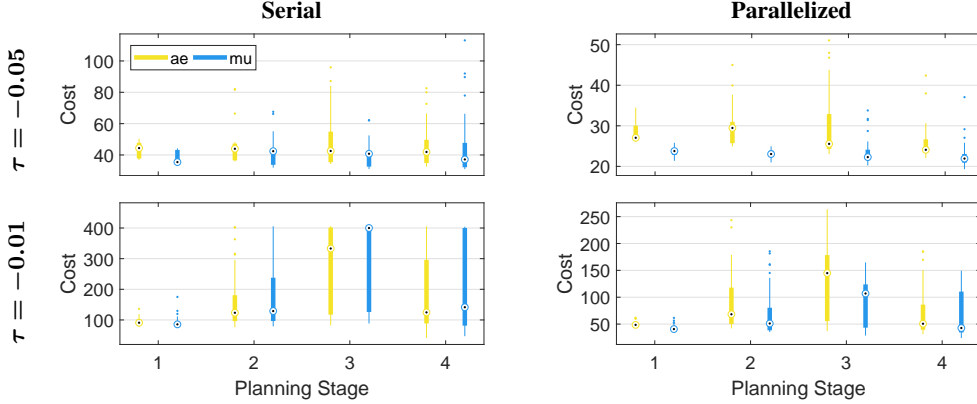

Figure 5: **Cost Vs Planning round.** Median and quartiles from 50 random trials of costs realized for rounds 1-4 of planning using serial (*left*) or parallelized (*right*) sampler for target performance of $-0.05$ (*top*) and $-0.01$ (*bottom*). `bed-lb` always utilizes the *full* budget, $C_{\max} = 400$, whereas the iterative approaches `ae` and `mu` utilize (sometimes significantly) less when the problem structure and targeted performance allows for it.

The iterative approaches `ae` and `mu` often utilize less than the max budget of $400$ as seen in Fig. 5 to achieve regret-style guarantees of $\tau = -0.05, -0.01$ whereas `bed-lb` always utilizes the full budget. Despite sometimes significant reduction in computation (*e.g.* parallelized, with $\tau = -0.01$), both iterative approaches achieve performance comparable to full-budget `bed-lb`, seen in Fig. 6. Over half the designs do not receive additional evaluation after the initial allocation of computation to obtain loose bounds under `mu`. Though it is possible to set a different budget wherein `bed-lb` uses comparable computation as the refinement algorithms, this illustrates the challenges of identifying a meaningful computation budget without knowledge or analysis of the unknown problem structure. We find that `mu` yields computational savings over `ae` in the parallelized sampler. Both perform similarly in the serial sampler where posterior inference is exceedingly expensive.

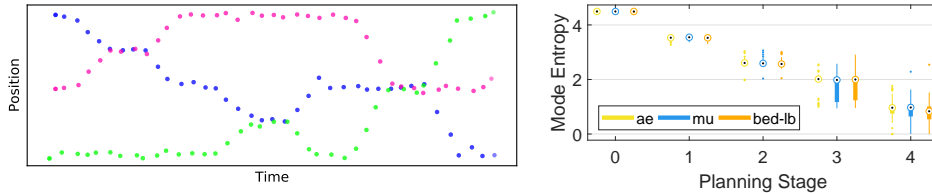

Figure 6: *Left:* Multi-object tracking scenario where $K = 3$ targets repeatedly become proximate, causing tracking ambiguities. *Right:* Median and quartiles of the entropy of the empirical distribution of posterior modes from 50 random trials following each BOED design stage. Empirical mode distribution is estimated by mapping each of 3k samples to 24 known posterior modes. Reduction in mode entropy indicates resolution of track ambiguities arising from entanglements.

## 5 Conclusion

Through iterative refinement of information bounds we have shown that BOEDIR can adaptively allocate limited computational resources to efficiently identify high-quality designs. We have chosen to focus on simple nested Monte Carlo MI bounds for clarity, but note that BOEDIR can use any MI bounds that can be iteratively refined. Options include sample-based [29, 24], purely variational [22], or a hybrid bounds [11, 12, 7]. Nevertheless, using simple bounds we find that our adaptive refinement procedure achieves significant computational savings through cost-aware allocation of resources and the provision of guarantees that allows for relaxing of the desired performance. Furthermore, in problem structures where there are few highly informative rewards but many poor ones (akin to the *scarce* motif), BOEDIR can provide the greatest benefit while minimizing computational effort.

## Broader Impacts

Despite its promise and longstanding research focus, BOED has seen limited practical utility due to the difficulties associated with evaluating information measures. Nevertheless, practical algorithms for effective information retrieval, which reason about uncertainty in a Bayesian context, have widespread value. This work posits that by accounting for identifiable cost structure, and the judicious allocation of resources, one can integrate Bayesian reasoning to experimental design in a practical way. Our approach utilizes readily available bounds and places minimal assumptions on model complexity. Having emphasized the positive aspects of this work, we acknowledge limitations of the approach. In particular, we assume that a cost structure is known or otherwise easily estimated. In practice we find that empirical estimates of costs are easily obtained, but acknowledge that this may not be true for all cases. Despite any limitations, our approach is broadly applicable and we therefore expect this work to have significant impact on the practical application of BOED in a wide range of settings.

## Acknowledgments and Disclosure of Funding

This research was partially supported by ONR (Award No. N00014-17-1-2072) and DOE/NNSA (Award No. DE-NA0003921).

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
