[Supplementary Material]

# Supplementary Material: Sequential Bayesian Experimental Design with Variable Cost Structure

**Sue Zheng**
CSAIL MIT
szheng@csail.mit.edu

**David S. Hayden**
CSAIL MIT
dshayden@csail.mit.edu

**Jason Pacheco**
University of Arizona
pachecoj@cs.arizona.edu

**John W. Fisher III**
CSAIL MIT
fisher@csail.mit.edu

## 1 Bounds at Subsequent Iterations

The main paper provides a brief overview of nested Monte Carlo bounds. We give a more detailed derivation of those specific bounds here and discuss alternative bounds in Sec. 3. We begin with notation elements that are used throughout. Let $\mathcal{D}_r$ be the outcomes of past experiments at stage $r$. We further assume samples of the latent variables $\mathcal{X} = \{x_n\}_{n=1:N}$, drawn from updated beliefs, $x_n \sim p(x \mid \mathcal{D}_r)$, and data samples $\mathcal{Y} = \{y_n\}_{n=1:N}$, drawn from the forward model of the candidate experiment, $y_n \sim p_a(y \mid x_n)$. We can obtain upper and lower bounds at subsequent stage $r$ as follows.

### 1.1 Upper bound on MI

By Jensen's inequality, we can lower bound log-evidence $\log p(y \mid \mathcal{D}_r)$ through any unbiased estimator of $p(y \mid \mathcal{D}_r)$,

$$\log p(y \mid \mathcal{D}_r) = \log \mathbb{E}[\hat{p}(y \mid \mathcal{D}_r)] \geq \mathbb{E}\left[\log \hat{p}(y \mid \mathcal{D}_r)\right]. \tag{1}$$

A simple choice is a Monte Carlo estimate: $\hat{p}(y \mid \mathcal{D}_r) = \frac{1}{M} \sum_{j=1}^{M} p(y \mid x_j)$ using samples $x_j \sim p(x \mid \mathcal{D}_r)$. Directly plugging the bound on log-evidence in Eqn. (1) into the expression for MI yields an upper bound:

$$I(X;Y \mid \mathcal{D}_r) = \mathbb{E}\left[\log \frac{p(y \mid x_0)}{p(y \mid \mathcal{D}_r)}\right] \leq \mathbb{E}\left[\log \frac{p(y \mid x_0)}{\frac{1}{M}\sum_{j=1}^{M} p(y \mid x_j)}\right] \triangleq \mathcal{U} \tag{2}$$

where the expectation is with respect to the joint distribution $x_0, y \sim p(x, y \mid \mathcal{D}_r)$ and marginal distribution $x_j \sim p(x \mid \mathcal{D}_r)$. We can readily estimate the bound using existing samples, as $x_n, y_n$ are drawn from the joint and $x_{j \neq n}$ arises from the marginal,

$$u = \frac{1}{N} \sum_{n=1}^{N} \log \frac{p(y_n \mid x_n)}{\frac{1}{N-1}\sum_{j \neq n} p(y_n \mid x_j)}. \tag{3}$$

### 1.2 Lower bound on MI

We can upper bound log-evidence $\log p_a(y \mid \mathcal{D}_r)$ using the estimator proposed by [7],

$$\log p(y \mid \mathcal{D}_r) \leq \mathbb{E}\left[\log \frac{1}{M+1}\sum_{j=0}^{M} p(y \mid x_j)\right] \tag{4}$$

where $x_0 \sim p(x \mid y, \mathcal{D}_r)$ and $x_{1:M} \sim p(x, \mathcal{D}_r)$. We can bound $\log p(y_n \mid \mathcal{D}_r)$ using existing samples by noting $x_n \sim p(x \mid y_n, \mathcal{D}_r)$ and $x_{\setminus n} \sim p(x, \mathcal{D}_r)$. Plugging the upper bound on log-evidence from Eqn. (4) into the expression for MI yields the following lower bound

$$I(X; Y \mid \mathcal{D}_r) \geq \mathbb{E}\left[\log \frac{p(y \mid x_0)}{\frac{1}{M+1} \sum_{j=0}^{M} p(y \mid x_j)}\right] \triangleq \mathcal{L} \tag{5}$$

where $x_0 \sim p(x \mid \mathcal{D}_r), y \sim p(y \mid x_0)$ and $x_{1:M} \sim p(x \mid \mathcal{D}_r)$. We can estimate the bound from samples as,

$$l = \frac{1}{N} \sum_{n=1}^{N} \log \frac{p(y_n \mid x_n)}{\frac{1}{N} \sum_{j=1}^{N} p(y_n \mid x_j)} \tag{6}$$

where $x_n \sim p(x \mid \mathcal{D}_r)$ and $y_n \sim p(y \mid x_n)$.

## 2 Practical Considerations for Using Bound Estimates

While the bounds holds in expectation, unbiased estimates of the bounds may not hold. Practically, we recommend adding a multiple of the empirical standard deviation to the upper (and subtracting from the lower) bound estimate to account for the variability arising from estimation of the bounds, *e.g.*

$$\hat{u} = u + 2\sigma_u \tag{7}$$

where we use two standard deviations in our experiments. The standard deviation can be estimated by noting the bound estimate is the average of $N$ samples, each with sample variance $\sigma_{u,1}^2$. Thus, when i.i.d. samples are used, the estimator variance is given as $\sigma_u^2 = \frac{1}{N}\sigma_{u,1}^2$. We estimate the single sample variance with $\sigma_{u,1}^2 = \frac{1}{N-1} \sum_{n=1}^{N}((u_a)_n - \bar{u}_a)^2$ where $(u_a)_n \triangleq \log \frac{p(y_n|x_n)}{\frac{1}{N-1} \sum_{m \neq n} p(y_n|x_m)}$ and $\bar{u}_a \triangleq \frac{1}{N} \sum_{n=1}^{N}(u_a)_n$. The above derivations can be readily extrapolated to obtain the empirical lower bound standard deviation $\sigma_l$.

## 3 Alternative Bounds in BOEDIR

While the analysis and experiments in the main discussion of Bayesian Optimal Experimental Design with Iterative Refinement (BOEDIR) make use of specific sample-based bounds, the framework is compatible with any bounds that support iterative refinement. As stated in the introduction (and elsewhere), there are alternatives, one of which we expound upon now. There exist several forms of upper and lower bounds, including purely sample-based bounds [9, 6], purely variational bounds [4], or hybrid bounds that fit a variational distribution to samples from the target distribution [2, 3]. Mechanisms for refinement vary depending on the form of the bound; drawing additional samples may be used to refine both sample-based and hybrid bounds while performing additional steps of gradient descent or broadening the variational class may refine the hybrid and variational forms. We discuss possible refinement mechanisms and costs associated with hybrid bounds next.

### 3.1 Hybrid Sample-Variational Bounds on MI

For brevity, we drop reference to the dataset $\mathcal{D}$ of observed outcomes from previous rounds of experimental design in sequential BOED but its impact on these bounds is similar to that for purely sample-based bounds discussed previously in Sec. 1. For the lower bound, we consider the adaptive contrastive estimate (ACE) of expected information gain developed in [3]. It is defined as,

$$\mathcal{L}_{ACE} = \mathbb{E}\left[\log \frac{p(y \mid x_0)}{\frac{1}{M+1} \sum_{m=0}^{M} \frac{p(x_m)p(y|x_m)}{q_\phi(x_m|y)}}\right] \leq I(X; Y) \tag{8}$$

where $x_0 \sim p(x \mid y)$, $x_{1:M} \sim q_\phi(x \mid y)$ and the expectation is taken over $\{y, x_{0:M}\} \sim p(y, x_0) \prod_{m=1}^{M} q_\phi(x_m \mid y)$. The variational distribution $q_\phi$ with variational parameters $\phi$ approximates $p(x \mid y)$. This lower bound is increasing with $M$ and becomes arbitrarily tight as $M \to \infty$,

making it amenable to iterative refinement through additional samples from $q_\phi$. Since the expectation is often not available in closed form, an unbiased estimate is obtained as,

$$l_{ACE} = \frac{1}{N} \sum_{n=1}^{N} \log \frac{p(y_n \mid x_{n,0})}{\frac{1}{M+1} \sum_{m=0}^{M} \frac{p(x_{n,m})p(y_n|x_{n,m})}{q_\phi(x_{n,m}|y_n)}} \qquad (9)$$

where $(y_n, x_{n,0}) \sim p(y, x)$ and $\{x_{n,m}\}_{m=1}^{M} \sim q_\phi(x \mid y_n)$.

For the upper bound, we consider the variational nested Monte Carlo (VNMC) estimator from [2] given as,

$$\mathcal{U}_{VNMC} = \mathbb{E}\left[ \log \frac{p(y \mid x_0)}{\frac{1}{M} \sum_{m=1}^{M} \frac{p(x_m)p(y|x_m)}{q_\phi(x_m|y)}} \right] \geq I(X;Y) \qquad (10)$$

where $x_0 \sim p(x \mid y)$, $x_{1:M} \sim q_\nu(x \mid y)$ and the expectation is taken over $\{y, x_{0:M}\} \sim p(y, x_0) \prod_{m=1}^{M} q_\phi(x_m \mid y)$. The variational distribution $q_\nu$ is again an approximation to the posterior $p(x \mid y)$. This bound also lends itself to iterative refinement as it decreases with $M$ and becomes tight as $M \to \infty$. An unbiased estimate is obtained as,

$$u_{VNMC} = \frac{1}{N} \sum_{n=1}^{N} \log \frac{p(y_n \mid x_{n,0})}{\frac{1}{M} \sum_{m=1}^{M} \frac{p(x_{n,m})p(y_n|x_{n,m})}{q_\phi(x_{n,m}|y_n)}} \qquad (11)$$

where $(y_n, x_{n,0}) \sim p(y, x)$ and $\{x_{n,m}\}_{m=1}^{M} \sim q_\phi(x \mid y_n)$.

Note that little additional computation is necessary to obtain *both* upper and lower bounds. The bounds differ by a *single sample* in the denominator and both variational distributions $q_\phi, q_\nu$ are approximations to the same distribution $p(x \mid y)$, allowing a single variational distribution to be learned and used for both bounds, $q_\phi = q_\nu$.

Rather than learning a separate approximation to $p(x \mid y_n)$ for each sample $y_n$, both works use the ideas of amortized variational inference to learn an inference network that takes in $y_n$ and outputs distribution $q_\phi(x \mid y_n)$. The inference network can be learned using stochastic gradient descent with computational cost $\mathcal{O}(N + M)$.

## 3.2 Refinement Mechanisms and Costs

The bound estimates in Eqns. (9) and (11) are consistent as both $M, N \to \infty$. Increasing $N$ reduces the variance of the bound estimator whereas increasing $M$ reduces the bias. When the bound estimate accounts for the variance of the estimate as in Eqn. (7), increasing either sample count serves to decrease the upper bound $u_{VNMC}$ and increase the lower bound $l_{ACE}$. We now outline the costs associated with these different refinement mechanisms. Note that both mechanisms can be included as options in the framework; our algorithm (MU) would assign different value and costs to each refinement option and weigh them accordingly to select a refinement procedure on a design.

Suppose we maintain a set of joint posterior samples $\mathcal{X}_0$, observation samples $\mathcal{Y}_a$ and variational samples $\mathcal{X}_{a,1:M_a}$. Let $N_a = |\mathcal{Y}_a|$. Refinement of design $a$ through increasing $N_a$ by $\eta$ samples $(N'_a = N_a + \eta)$ requires $\eta$ additional observation samples to be drawn, each with cost $c_y$. Posterior samples, each with cost $c_p$, need only be drawn if the updated observation sample counts exceed the existing posterior sample count, $N'_a \geq |\mathcal{X}|$. An additional $M_a$ variational samples must be drawn *per new observation sample* $y_n$: $\{x_{n,m}\}_{m=1}^{M_a} \sim q_\phi(x_{n,m} \mid y_n)$, each with cost $c_v$. Note that possibly many variational samples must be drawn, but each is often considerably less computationally expensive than posterior sampling, $c_v \ll c_p$; however in aggregate it can become comparable for $M_a$ sufficiently large. Only $\mathcal{X}_0$ is shared across designs and acts as overhead cost. Lastly the bound evaluation itself is bilinear in $N, M$ with cost $c_{\text{bound}}(N'_a, M_a) = d_0 + d_1 N'_a + d_2 M_a + d_3 N'_a M_a$. The net cost for refining $a$ by increasing $N$ is $c_a^N = \eta c_p * \mathbb{1}_{N'_a \geq |\mathcal{X}|} + \eta M_a c_v + \eta c_y + c_{\text{bound}}(N'_a, M_a)$.

If we consider increasing $M_a$ instead of $N_a$, we do not require any new observation samples nor posterior samples. We incur the cost of drawing $\eta N_a$ new variational samples and evaluating the bound. The net cost for refining $a$ by increasing $M$ is then $c_a^M = \eta N_a c_v + c_{\text{bound}}(N_a, M'_a)$.

When the bound estimate is dominated by the variance term in Eqn. 7, the gain from increasing $M$ is reduced relative to increasing $N$ through the estimated change to lower and upper bounds under

each refinement procedure. Conversely, when the bound estimate is dominated by the bias term, the gain from increasing $M$ is increased relative to $N$. Our approach can incorporate these differences in costs and expected improvements to the guarantee $g^*$ across refinement procedures as well as across arms to select refinements that have high expected marginal utility.

## 4    Knapsack Problem and Heuristic

The main paper referenced our marginal utility (MU) being an adaptation of the greedy knapsack heuristic. We now make that relationship explicit.

Let $\Omega$ be the set of all items, where each item $i$ has value $v_i$ and weight $w_i$. The $0-1$ knapsack problem maximizes over subsets $t$ the total value of items subject to a weight (budget) constraint $W$:

$$\max_{t \in \{0,1\}^{|\Omega|}} \sum_{i=1}^{|\Omega|} t_i v_i$$

$$\text{s.t.} \sum_{i=1}^{|\Omega|} t_i w_i \leq W$$

This optimization problem is known to be NP-hard [5]. A simple greedy approximation algorithm iteratively adds the item with highest value-per-unit-weight (marginal utility) until the weight limit is reached. Note that this greedy approach is optimal for the 'fractional' knapsack problem wherein items can be broken up arbitrarily to include a fraction of its weight and value.

For our setting, we would like to identify designs (items) that, when refined, will yield the greatest improvement in the highest performance guarantee $g^*$ (net value). To apply greedy knapsack, we need to specify costs and values for each item. The refinement of each design has a known cost, $c_a$ but unknown, random value. As discussed in the main paper, we propose to assign value based on an estimated change to $g^*$.

**Relation to knapsack problem**    While we use the greedy knapsack heuristic of selecting items with high cost-to-weight ratio, our problem does not naturally correspond to the typical knapsack formulation as the item cost and value depends on the items already included in the knapsack (selected for refinement) and their observed outcomes (updated lower and upper bounds). The cost for refining design $k$ may be reduced if prior inclusion of design $j$ incurred the overhead cost of posterior sampling. The value of refining the design with second highest upper bound has value dependent on the upper bounds of other designs and is increased if the highest upper bound has previously been refined and its updated upper bound decreases significantly.

**Relation to existing bandits problems with knapsack policies**    There exists work in the bandits literature that considers variable costs for each arm with budget constraints and incorporate a knapsack policy [1, 8]. However, these works consider the cumulative regret setting whereas our problem is concerned with the performance/reward of a final selection which is better represented as *simple* regret. Furthermore, as mentioned in the main paper, our problem differs from standard bandit problems altogether as *unbiased* observations of the rewards are not available.

## 5    Additional Gaussian Experiment Results

We give additional comparisons against BOED baseline and against alternative refinement algorithms within BOEDIR.

### 5.1    Comparison against BOED baselines: Additional budgets and relative costs

We compare our approach against two proxies `bed-lb`, `bed-ub` in the standard proxy-based BOED framework. Fig. 1 presents the results for various relative posterior costs, budgets, and the three motifs discussed in the main paper. Our approach often achieves comparable performance to the better of the proxies while *also* providing a performance guarantee.

Figure 1: **Performance Against Proxy-based Baseline for Various Relative Posterior Costs.** Median and quartiles from 50 random trials of realized performance for three motifs. We compare our approach (`mu`) against two proxies, `bed-lb` and `bed-ub`, in the standard BOED framework subject to the same budgets. *Top-bottom:* Relative posterior sampling cost varies from [0.1, 1, 10, 100] to that of observation sampling. Budgets $C_{\max}$ of [250, 250, 500, 1000] *(left-column)* and [500, 500, 1000, 2000] *(right-column)* for the respective posterior costs.

## 5.2 Comparison against alternative refinement algorithms: Performance guarantees

Fig. 2 shows the performance guarantees provided when iteratively refining bounds subject to a maximum budget. The resulting trends are very similar to the case wherein we examine the cost subject to a target performance guarantee.

As before, with *scarce* high rewards (top row of Fig. 2), all algorithms for refinement selection perform similarly as few designs merit refinement. As the distribution of MI provides increasing number of designs meriting consideration, it becomes increasingly challenging to provide a performance guarantee as it requires sufficient refinement of the bounds for those designs. This is evident in the decreasing guarantee *broad* and even more so with *abundant*. However, we find that our approach `mu` allocates resources in a cost-effective way to achieve higher performance guarantees when subject to the same budget.

As the relative posterior costs (overhead) increases (left to right) in the plots, we find the two baselines, `ae` and `lucb`, swap in terms of their relative performance. `lucb` performs better when overhead costs are negligible whereas `ae` performs when overhead is especially costly. We find that our approach, taking into account these costs, performs equal (in *scarce*) or better than the two baseline across cost structures.

Figure 2: **Performance Guarantee Vs Budget as Overhead Cost Increases for Three Motifs.** Median and quartiles from 50 random trials of performance guarantees at each budget level. Posterior sampling costs vary from [0.1, 1, 10, 100] (*left-right*) to that of observation sampling. Motifs are: *scarce* mostly uninformative (*top-row*), *broad* similarly informative (*middle-row*), and *abundant* mostly informative (*bottom-row*). Baselines `ae` and `lucb` perform well at low and high posterior costs respectively whereas `mu` performs well across cost ratios. All require equally little computation in *scarce* as loose bounds suffice to identify many designs as poor. As more designs yield high information, additional computation is required to refine bounds sufficiently to achieve the target performance guarantee.

# 6   Description of Tracking Model in Experimental Results

The generative model for multi-object tracking is given as

$$p(z) \propto \mathcal{I}(z), \; p(x \mid z) = \prod_{k,t} \mathcal{N}\left(x_{tk} \mid f(x_{(t-1)k}), Q\right), \; p(s \mid x, z) = \prod_{n,t} \mathcal{N}\left(s_{tn} \mid h(x_{tz_{tn}}), E\right)$$

where $x_{tk}$ is the latent state for target $k$ at time $t$, $s_{tn}$ is the $n^{\text{th}}$ sensor observation at $t$, $z_{tn} = k$ is the association of $s_{tn}$ to target $k$, and $Q, E$ are dynamics and sensor covariances. $\mathcal{I}(z)$ is the constraint that each target receives exactly one association at each time $t$. We use linear dynamics and sensor models $f, h$.

Incorporation of sensor data results in a multi-modal posterior $p(x \mid s)$ that we sample from. In Fig. 3 we illustrate four different assignments of sensor observations to targets, each corresponding to a different mode in the posterior (out of 24 possible modes). Each round $r$ of BOED seeks to identify an annotation that maximizes information about the target state, $I_a(x; y \mid s)$. Each design $a$ maps to the specification of two sensor observations $s_{t_1 n_1}, s_{t_2 n_2}$. The annotator reports whether these sensor data arise from the same underlying object with error probability $p_{\text{annot}}$: $p_{a((t_1,n_1),(t_2,n_2))}(y \mid z) = p_{\text{annot}}$ for $y = \mathbb{1}(z_{t_1 n_1} = z_{t_2 n_2})$. Useful annotations resolve track ambiguities to reduce the number of modes remaining in the posterior.

Figure 3: **Multiple Modes in Target Posterior.** Four different matches of entering to exiting objects at 'entanglement' events. Each corresponds to a mode of the posterior over target states. Annotations can help resolve such ambiguities and reduce the number of modes.