[Reviews · NeurIPS 2020]

Review 1

Summary and Contributions: Post author response --------------------------- It seems there is a difference of a opinion on this paper, and I still prefer to push for acceptance. On the question of whether they should compare to variational bounds, I feel this is slightly orthogonal to this problem and, whilst it’s not in the main paper, is discussed in Appendix 3. Variational bounds wouldn’t be straight up baselines for the method proposed here, instead you could use the variational bound pair ACE/VNMC in place of the non-variational bound pair PCE/NMC within the method of this paper. I’m not convinced that the method presented is limited to 1D designs, but rather to finite design spaces. I agree with R3 that the bounds should be introduced with citation, so it’s clear where they are coming from. I personally disagree with R4 about the level of technical novelty, and think it’s sufficient. I agree with other reviewers’ comments about readability and clarity, and think this is the biggest weakness. -------------------------------------------------------------------------------------- Performing Bayesian optimal experimental design (BOED) with a MI objective is highly desirable, but computationally challenging. In a discrete design space setting, resources can be allocated between designs progressively to refine the most promising designs and this paper proposes a new cost-aware framework to do this resource allocation. Using existing upper and lower bounds on MI, the authors formulate regret bounds on the design choice. The authors then estimate the marginal utility (MU) of using additional resources on any particular design and use this to guide resource allocation. They compare this new method with 1) methods taking the upper or lower bounds as a proxy objective, 2) other resource allocation methods using ideas from MABs. In experiments, this is method is shown to perform about as well as taking the lower bound as a proxy (bed-lb) and better than other methods.

Strengths: - This paper tackles an important problem. When using BOED in a large but discrete design space, correctly allocating resources is very relevant particularly in a setting with “scarce” information (as in the example of this paper) - By considering regret bounds, the authors provide a more well-founded criterion than picking a lower or upper bound as a proxy objective - The authors show that these existing lower and upper bounds can be used to define a resource allocation method. Using two bounds, as opposed to the unbiased estimates expected in conventional MABs is a novelty that is well-suited to this problem - This work suggests new directions for using upper and lower bounds in the resource allocation problem in BOED and beyond - The level of experimentation is appropriate and indicates that the new method (MU) does as well as or better than the baselines. I believe the authors have done a good job of including reasonable baselines that allow us to fairly evaluate their proposed method.

Weaknesses: - I believe clarity may be an issue for this paper. For instance, in the section 104-110 the authors seem to introduce the notation g(I_a,I_a^*) without defining it. I believe I understand that g will, in fact, be one of g_regret and g_pct but as these are introduced later it made for difficult reading. It could be worth spelling out exactly what “refinement” is (drawing additional samples, but how many in one go, etc.). In lines 134-140, the approximates come thick and fast. It could be worth including some discussion of why these approximations are made (e.g. unless I’ve missed something we could actually different the max function unless we’re sitting right on a discontinuity point) and how much they might impact the final algorithm. It’s just a suggestion but the authors might consider relegating some details of the GMRF to the appendix and expending more ink on the core of their approach, especially around 133-140. - It should be explicitly stated at line 82 that these bounds have been considered in prior work. E.g. the lower bound appears in reference [11] as PCE and the upper bound has appeared in several existing works (see [10] for example). - It’s a pity that `bed-lb` appears to equal or outpeform `mu`. The paper would be even stronger if a practical case where this doesn’t happen could be demonstrated. That said, I believe the paper stands without this additional experiment.

Correctness: The methodology is correct and clever. As mentioned, some approximations around lines 133-140 could be further explained.

Clarity: I believe clarity is a weakness of this paper and outlined some specifics in ‘weaknesses’. That said, I was able to understand the paper, albeit with a bit of reading backwards and forwards.

Relation to Prior Work: As mentioned before, it should be explicitly stated with citations that the upper and lower bounds used have come from the literature. Also, I would recommend the authors check the paper ‘The DARC Toolbox: automated, flexible, and efficient delayed and risky choice experiments using Bayesian adaptive design’ by BT Vincent and T Rainforth, which considered a related but different approach to resource allocation using ideas from the bandit literature. That said, I believe the authors have done a good job of comparing to existing work especially in the appendix. The work is clearly distinct from existing approaches and the idea of using upper and lower bounds together provides a new direction for potential future work.

Reproducibility: Yes

Additional Feedback: I enjoyed reading this paper and I believe it makes a notable additional to the field of Bayesian experimental design. Further clarity in the writing would be the number one place for further improvement to allow a wider audience to appreciate the ideas of the paper.


Review 2

Summary and Contributions: The work proposes to iteratively refine a Monte Carlo based Mutual Information bound, while considering cost of sampling to refine the bound.

Strengths: The proposed idea seems to be interesting for a small scale optimal experimental design problem, where one can trade between allowed cost and gained information with a relatively clear write-up.

Weaknesses: The main weakness of the method is the baseline the approach compares itself with. It would have been very interesting to see how the approach compares with recent variational based approaches.

Correctness: Formulation for the most part seems correct (did not check thoroughly).

Clarity: In general, yes.

Relation to Prior Work: Previous works are for the most part included and discussed. However, the the problem the proposed method sets out to solve is not compared with a recent approach that might do as good a job in a unified and scalable way-- this comment is included in the feedback section.

Reproducibility: Yes

Additional Feedback: Major comments: 1) Doesn’t the Monte Carlo estimates, Eq(4), need to be computed for each experimental design setup? One would also need to compute the expected information gain, requiring a nested Estimator, given Eq(2), right? Given this, could you please elaborate on the choice of the nested MC estimator as opposed to , e.g., variational approach? Reference. Tom Rainforth et.al., “On Nesting Monte Carlo Estimators”, ICML 2018. 2) What is the added computational cost of the knapsack algorithm to the proposed method, and how does it scale to experiment setup with lots of variables? 3) How does the method compare (in terms of cost, scalability, and guarantee) with the recent method of Foster et.al., “A Unified Stochastic Gradient Approach to Design a Bayesian-optimal Experiments”,AISTATS 2020? 4) The motivation behind the exact cost function formulation (line 125) is not clear. It in fact seems very ad-hoc, is this right? Could you please elaborate? Minor comments: 1) Neither the caption of Figure 1 nor the line referring to it (line 94) explains the experimental scenario or what the figures depict adequately.


Review 3

Summary and Contributions: The authors propose a novel Bayesian optimal experimental design framework that takes into account the varying computational cost of evaluating the utility function at a particular experimental design. Their method iteratively refines a lower and upper bound on the mutual information over the design domain, where the bounds are refined, e.g. via more samples, in domain regions where the trade-off between potential performance improvement and computational cost is ‘best’. Their experiments show that their method requires lower or equal cost to baseline methods for a variety of different experimental budgets, while often outperforming them.

Strengths: This work neatly tackles the issue of varying computational cost of the utility function in Bayesian optimal experimental design. Furthermore, they iteratively refine estimates of upper and lower bounds on the mutual information utility. This means that they only need to tighten the bounds at experimental designs that matter, i.e. have a high prospect of improving upon the current best design or have a very low computational cost. By using a greedy knapsack algorithm to facilitate this, they reduce computational resources as much as possible, which is reflected in their experimental findings. Their experiments are thorough and detailed. They make many interesting comparisons of how their method compares to baselines in different experimental budget scenarios; moreover, their analyses are sound and intuitive.

Weaknesses: The authors only considered one-dimensional experimental designs. More often than not, however, experimental designs are higher dimensional than that in practice. This undermines the applicability of their method. At minimum, I would expect a discussion on scalability to higher design dimensions and how they expect their method to fair in that scenario. While not peer-reviewed, there is some recent work on non-myopic sequential BOED that might be worth comparing to: Jiang, S., Chai, H., González, J., & Garnett, R. (2020). BINOCULARS for Efficient, Nonmyopic Sequential Experimental Design. arXiv: Learning. The authors considered one toy experiment and one realistic experiment. While these two experiments were covered well, I would have liked to see another realistic experiments such as the multi-object tracking problem. I realise that space and time might be an issue, but this would significantly highlight the advantages of their method. ***EDIT***: I believe the authors cleared up the issue with the scalability w.r.t. design dimensions in their author response. I agree that for discrete design this concern becomes practically irrelevant, which, including other points, leads me to increase my score to an accept. I would hope, however, that the authors clarify the fact that they consider discrete designs at the end of their introduction.

Correctness: There does not seem to be anything incorrect with the methodology.

Clarity: The paper is generally well-structured and well-written, but there are a few grammatical mistakes. I would suggest the authors to proof-read their submission again. There are also a few areas that lack clarity of exposition. While the maths was mostly clear, I would have liked to see a derivation of the upper and lower bounds in Equation 4, and/or a proof that these bounds are valid (either through a reference or through their own doing). The authors attempt to do derive both bounds in section 1.1 and 1.2 in the supplementary material, but leave out steps that I believe would help the reader understand the bounds more effectively (specifically, arriving at Equations 1 and 3 of the supplementary material). Equation 4 is critical for the remainder of the paper, so it is important for the reader to understand where it is coming from. The authors don’t refer to Algorithm 1 on page 2 at all in the main text. This might make the reader feel like they have missed something in the text. I would suggest to add a short sentence in the introduction of Section 2 to rectify this. In line 126/127 of page 4 the authors claim that the parameters of the quadratic cost function are “usually readily estimated”. I would like to know exactly how these are usually estimated (in the method section, or for their specific experiments) because, as the authors noted, these parameters highly depend on the problem in question and are paramount to their method. I have one last note: In line 71-73 of page 2 the authors argue that the sequential optimization shown in equation 2 “avoids the exponential complexity associated with optimising over all possible designs”. First, they are right that by considering myopic sequential Bayesian experimental design (BED), as opposed to non-myopic static BED, they are reducing the computational burden. This somehow suggests that this option would always be preferred to having to determine all optimal designs a^\ast_{1:T} jointly. However, they are forgetting that from an information-theoretic point of view myopic sequential BED is less informative than non-myopic static BED (as for the latter you are trying to jointly determine all optimal designs a^\ast_{1:T}, i.e. ‘looking further into the future’). This is not a negative point of their method, as sequential and static BED have very different use cases. It is important, however, to make that difference clear. Second, the statement of “over all possible designs” makes it sound like they have avoided the optimisation over the whole design domain. Instead, however, they mean that they do not have to find a^\ast{1:T} jointly, which is very different. I feel like this could be expressed more clearly. ***EDIT***: The authors seem to have responded to all of the above concerns and promised to address them in their revision.

Relation to Prior Work: In section 3.2 the authors describe the relation and difference of their method to best-arm identification problems for multi-armed bandits. Since the authors talked about computational costs associated with each design, it would make sense to also briefly talk about other, external sources of cost. Many real-life problems have an inherent cost associated for each design, such as, for instance, larger designs costing more money. Briefly discussing this scenario would help set the author’s method apart from this research field, which also tries to combine maximising a utility function with minimising some form of (other) cost. Examples that I can think of are: Palmer, J. & Müller, P. (1998). Bayesian optimal design in population models for haematologic data. Stat. Med., 17, 1613–1622.
 Stroud, J., Müller, P. & Rosner, G. (2001). Optimal sampling times in population pharmacokinetic studies. J. R. Stat. Soc. Ser. C. Appl. Stat., 50(3), 345–359

Reproducibility: Yes

Additional Feedback: While Algorithm 2 is written clearly enough, I would ask the authors to kindly provide research code that can be used by others to reproduce their results. In Figure 1, the x-axis label ‘MI’ in the top row is cut off by the bottom row plots. Also, the labels are slightly hard to read because of their small size.


Review 4

Summary and Contributions: The paper considers Bayesian optimal experimental design (BOED) with iterative refinement of mutual information (MI). The bounds of mutual information is calculated at every iteration, by which the guarantee of performance is derived. The design is selected if the guaranteed performance exceeds the target performance or the cost exceeds budget. The performance is verified through Gaussian MRF and object tracking.

Strengths: - The basic idea is interesting. - The procedure is not so complicated.

Weaknesses: - Readability is not good. - Empirical evaluation is not fully convincing. - Technical novelty is not outstandingly strong.

Correctness: I don't find any significant error.

Clarity: Readability can be improved.

Relation to Prior Work: Since the topic is not my expertise, for me, this is a bit difficult to evaluate. As an educated guess, I think related works are seemingly discussed adequately.

Reproducibility: Yes

Additional Feedback: The basic idea would be reasonable. Adaptively tightening the MI bounds is interesting. Since the comparison is provided only among a variant of proposed methods, it is difficult to evaluate practical usefulness of the proposed method. Other baselines with existing methods would be informative. The upper/lower bounds is only for 'expectation' of estimates. The reliability of the bounds is difficult to see, particularly when the sample is small. I felt difficulty to understand the paper mainly because sometimes explanations of words are delayed without clarifying it (e.g., refinement set, the function 'Refine' in Alg2, realized performance (Fig1), nominal updates, ...). At the end of 'Costs for sample-based bounds', the authors simply said several parameters can be readily estimated. More detailed description would be desirable. How is the nominal update Delta u = gamma_a,u*(u_a - l_a) is justified? Can the moving average really provide a good approximation? The selection of the refinement set should be described in the main sections. Currently, readers cannot find it until experimental section. I first couldn't find it when I read 'Algorithm 2'. In my understanding, this procedure is quite essential. If possible, providing more detailed discussion can be insightful for understanding the behavior of the proposed method. The horizontal axes of the top-row plots in Figure1 are not shown (overlaid by the bottom plots). Minor: Title is different between CMT and PDF. CMT:Sequential Bayesian Experimental Design with Variable Cost Structure PDF:Bayesian Experimental Design with Variable Costs

[Author Response · NeurIPS 2020]

We thank the reviewers for their insightful feedback. We are encouraged that they found our approach to be interesting
(R2, R4) and distinct from existing approaches (R1) with thorough and detailed experiments (R3) and reasonable
baselines for a fair evaluation (R1). We are pleased that R1 recognizes the novelty and value of using *both* upper
and lower bounds, *in contrast to existing approaches*, for resource allocation and for the provision of performance
guarantees (regret) that current methods lack. Indeed, incorporating existing estimators of MI that are *biased in known*
*directions* as bounds (rather than proxies for true MI) is the critical insight that directly leads to both algorithmic
improvement *and* performance guarantees.

(R2, R4) **Baselines** R1 feels the paper does "a good job of including reasonable baselines" while R2 and R4 prefer
comparison to additional MI bounds. We emphasize that our goal is not to identify the most accurate MI proxies, but to
propose an approach which exploits available bounds to guarantee performance with minimal computation. While we
consider specific bounds (Eqn. 4) **other bounds are easily substituted** including variational bounds suggested by R2
(Supplement Sec 3). R4 considers the comparisons to be "a variant of proposed methods", which we disagree with
since the typical Bayesian optimal experimental design (BOED) approach uses our chosen bounds ('bed-lb', 'bed-ub')
as proxies [5, 2, 3]. Additional comparisons suggested address a different, *continuous design*, problem (R2 [3], R3 [4]).

(R3) **Dimension Experiment or Discussion** R3 is concerned that the experiments are 1D designs. Design dimension
is relevant only for continuous designs, whereas in discrete settings the number of distinct design elements is a better
measure of complexity. For the Gaussian MRF we use a set size of 100 and for the Tracking experiment there are 6669
choices. We do not have experiments explicitly analyzing the impact of increasing set size, but expect our approach to
yield greater savings (w.r.t. baseline) as allocating resources to promising designs is increasingly important.

(R2) **One needs to compute the expected information gain (EIG), requiring a nested estimator, given Eq(2),**
**right?** Not exactly. The bounds (*e.g.* Eqn. 4 *are typically used as proxies for the true MI (Eqn. 2)*, but we explicitly
treat them as bounds. We select the design with the highest performance guarantee (L104-110) which is afforded by
two-sided bounds. We use nested estimators in our experiments because they are simple – and common in the BOED
literature [1, 3]. One could use (non-nested) alternatives (Supp. Sec 3), but we illustrate the benefits of an approach
incorporating two-sided bounds. We will also add references as suggested by R1, R3.

(R2) **Added cost of knapsack algorithm, scaling with variables** The algorithm only adds a small cost ($< .01\%$ of
the total computational cost for GMRF experiment) because the marginal utility (MU) of each design doesn't require
any computation over the samples. In general, cost of bound evaluation (quadratic in samples) will far outweigh that of
knapsack. The knapsack cost is *linear* in the number of designs since the MUs depend on each lower and upper bound.

(R2) **Motivation for the cost function formulation (L125) unclear.** The computational cost arises from the particular
bounds used in the BOEDIR framework. The nested bound evaluations in Eqn. (4) are quadratic in the total number of
samples ($= |\mathcal{Y}| + N$ where N is the incremental update when refining), resulting in the cost function of L125. The
costs take a different form for other bounds (Supp. Sec 3).

(R3, R4) **Estimating Costs** The costs for sampling can be directly estimated using any method for measuring code
performance, including functions that measure wall time. The coefficients of the bounding function can be estimated by
a quadratic fit to timing measurements at various sample sizes. Alternatively, one could learn these parameters online;
measuring and adaptively estimating the computational cost adds little computation.

(R2) **Does Eq(4) need to be computed for each experimental design setup?** Yes, we bound EIG of each design
(Eqn. 4) with an initial amount of computation. This may suffice to exclude some designs from further computational
resources; over half of the designs in the tracking experiment do not receive additional evaluation.

(R1) **Why are approximations in Sec. 3.1 made?** One can exactly evaluate the change in performance guarantee
under an assumed update to the lower/upper bounds. However, the result is sensitive to the update assumption due to
the discontinuous max function, so we use a standard smooth approximation: LogSumExp.

(R4) **Selection of the refinement set should be described.** The refinement set, $\mathcal{R}$ in Alg. 2, consists of all designs
with upper bound greater than the highest lowest bound. These are all designs that may feasibly be optimal.

(R1, R3, R4) **Presentation** In addition to comments above we will: clarify the definition of $g(I_a, I_a^*)$ (R1), give an
example of cost parameter estimation (R3, R4), explicitly reference Alg. 1 (R3), discuss suboptimality gap of greedy
(myopic) vs. non-myopic BOED (R3), and expand derivations of the bounds (Eqn. 4) in the supplement (R3). Additions
to the main text are minor, but we will shift details of the GMRF experiment to the supplement (R1) for extra space.

[1] H. Cho, B. Berger, and J. Peng. Reconstructing causal biological networks through active learning. *PloS One*, 11(3):e0150611, 2016.
[2] A. Foster, M. Jankowiak, E. Bingham, P. Horsfall, Y. W. Teh, T. Rainforth, and N. Goodman. Variational bayesian optimal experimental design. In *NeurIPS*. 2019.
[3] A. Foster, M. Jankowiak, M. O'Meara, Y. W. Teh, and T. Rainforth. A unified stochastic gradient approach to designing bayesian-optimal experiments. In *AISTATS*, volume 108. PMLR, 2020.
[4] S. Jiang, H. Chai, J. Gonzalez, and R. Garnett. Binoculars for efficient, nonmyopic sequential experimental design. *arXiv*, pages arXiv–1909, 2019.
[5] J. Pacheco and J. Fisher. Variational information planning for sequential decision making. In K. Chaudhuri and M. Sugiyama, editors, *PMLR*, volume 89 of *PMLR*. PMLR, 2019.


[Meta-Review · NeurIPS 2020]

The authors describe an interesting approach to using mutual information within Bayesian optimal experimental design that appears to compare favorably to existing methods with respect to computation. The main reviewer concern is clarity of presentation of the method and the specific contribution of the paper. Discussing what approximations are being employed and their potential consequences would make the work more compelling.